# Regulation and Functions of α6-Integrin (CD49f) in Cancer Biology

**DOI:** 10.3390/cancers15133466

**Published:** 2023-07-02

**Authors:** Rahele Khademi, Hossein Malekzadeh, Sara Bahrami, Najmaldin Saki, Reyhane Khademi, Luis G. Villa-Diaz

**Affiliations:** 1Systematic Review and Meta-Analysis Expert Group (SRMEG), Universal Scientific Education and Research Network (USERN), Tehran 1419733151, Iran; 2Immunology Board for Transplantation and Cell-Based Therapeutics (Immuno_TACT), Universal Scientific Education and Research Network (USERN), Tehran 1419733151, Iran; 3Department of Oral Medicine, Faculty of Dentistry, Ahvaz Jundishapur University of Medical Sciences, Ahvaz 6135715794, Iran; 4Resident of Restorative Dentistry, Qazvin University of Medical Sciences, Qazvin 3419759811, Iran; 5Thalassemia & Hemoglobinopathy Research Center, Health Research Institute, Ahvaz Jundishapur University of Medical Sciences, Ahvaz 6135715794, Iran; 6Department of Medical Laboratory Sciences, School of Para-Medicine, Ahvaz Jundishapour University of Medical Sciences, Ahvaz 6135715794, Iran; 7Department of Biological Sciences, Oakland University, Rochester, MI 48309, USA; 8Department of Bioengineering, Oakland University, Rochester, MI 48309, USA

**Keywords:** integrin, CD49f, ITGA6, molecular mechanisms, metastasis, resistance, stemness, epithelial-mesenchymal transition (EMT), hematological malignancies, diagnosis biomarker, target therapy

## Abstract

**Simple Summary:**

Integrins play key roles in mediating cell adhesion and delivering chemical and mechanical signals to the interior of the cell. Consequently, they actively control cellular proliferation, differentiation, and apoptosis. Integrin signaling dysregulation can be a major factor in the development of tumors, and it is involved in the processes of the malignant plasticity of the epithelium, the reactivation of metastasis, and resistance to therapeutic interventions. Here, we describe the current understanding of the α6-integrin subunit (ITGA6, also known as CD49f and/or VLA6; encoded by the gene *itga6*) in cancer cells. The roles of ITGA6 in cell adhesion, stemness, metastasis, angiogenesis, and drug resistance, and as a diagnostic biomarker, are discussed. The importance of ITGA6 in the progression of a number of cancers, including hematological malignancies, suggests its potential usage as a novel therapeutic target.

**Abstract:**

Over the past decades, our knowledge of integrins has evolved from being understood as simple cell surface adhesion molecules to receptors that have a complex range of intracellular and extracellular functions, such as delivering chemical and mechanical signals to cells. Consequently, they actively control cellular proliferation, differentiation, and apoptosis. Dysregulation of integrin signaling is a major factor in the development and progression of many tumors. Many reviews have covered the broader integrin family in molecular and cellular studies and its roles in diseases. Nevertheless, further understanding of the mechanisms specific to an individual subunit of different heterodimers is more useful. Thus, we describe the current understanding of and exploratory investigations on the α6-integrin subunit (CD49f, VLA6; encoded by the gene *itga6*) in normal and cancer cells. The roles of ITGA6 in cell adhesion, stemness, metastasis, angiogenesis, and drug resistance, and as a diagnosis biomarker, are discussed. The role of ITGA6 differs based on several features, such as cell background, cancer type, and post-transcriptional alterations. In addition, exosomal ITGA6 also implies metastatic organotropism. The importance of ITGA6 in the progression of a number of cancers, including hematological malignancies, suggests its potential usage as a novel prognostic or diagnostic marker and useful therapeutic target for better clinical outcomes.

## 1. Introduction

Integrins are a family of type I transmembrane glycoproteins ubiquitous in cells and play a key role in cell adhesion to extracellular matrix (ECM) proteins and to neighboring cells [1]. Integrins are a link between the extracellular content and the cell’s cytoskeleton and transmit bidirectional signals between cells and the ECM [2]. Upon their activation, signal transduction mechanisms are regulated and involved in gene expression, cell differentiation, motility, polarity, proliferation, survival, and apoptosis [3,4]. The integrin family is divided into 𝑎 and β subfamilies. The 𝑎 subfamily has 18 subunits, while the β subfamily has eight subunits. Specific 𝑎 subunits form heterodimer combinations with specific β subunits, forming 24 different heterodimer combinations with affinities for determined ECM components, such as collagens, fibronectin, and laminins. These integrin heterodimer combinations have specific cellular functions [1]. In this review, we focus our attention on the role and regulation of the 𝑎6 integrin subunit in cancer, as it has emerged as an important player in this malignancy. 

α6-integrin, also known as ITGA6, CD49f, or VLA-6, is a cell surface protein that mediates cell-to-cell and cell-to-stroma adhesion, which is important for cell proliferation, migration, survival, and differentiation [5]. In epithelial cells, it is also important for tissue architecture due to its preservation of the cell membrane integrity [6]. The *itga6* gene encodes the subunit of alpha integrin (α6) [7,8], which lacks an αA-domain [9]. With the exclusion of the greatly conserved membrane-proximal KXGFFKR motif, the cytoplasmic domains of α-integrins have few sequences in common [10]. ITGA6 dimerizes with either integrin β1 (ITGB1 also known as CD29) or β4 (ITGB4 also known as CD104) to form integrin α6β1 and α6β4, respectively [5,7,8,11]. In cells expressing both ITGB1 and ITGB4, ITGA6 appears to preferentially bind to ITGB4 [7]. Integrin α6β1 and α6β4 are highly selective laminin receptors [7,12]; however, recently, netrin4 was identified as a new ligand for α6β4 that, in large doses, inhibits the cell cycle in most cases, and its levels fall in cancer patients with glioblastoma [13]. 

ITGA6 is expressed from the early stages of development to adulthood and during the development of organs. Its overexpression in retinal pigment epithelia promotes proliferation, which is beneficial for the regeneration of injured tissues [13]. It also enhances and stabilizes Rpsa expression. PEDF-Rpsa-ITGA6 signaling is involved in neuronal morphogenesis [14]. Hematopoietic stem cells (Thy1(+)/Rho(low)/CD49f(+)) interact with the bone marrow microenvironment via ITGA6 [15,16]. In addition, it has been reported that immature hematopoietic stem cells in cord blood display high expression of ITGA6 [17]. Laminin-511, an ITGA6 ligand produced by human hematopoietic stem cells, is an important component of the stem cell niche that makes stem cells survive in the absence of other cytokines or growth factors [18]. Moreover, the expression levels of ITGA6 by thymus epithelial cells are relevant in the general control of the major histocompatibility complex (MHC) expression [19]. 

In tumor cells, ITGA6 expression and activity improperly change, promoting signaling pathways that govern carcinogenesis, invasion, angiogenesis, metastasis, immune evasion, stemness, and resistance [5,6,8,20]. For example, ITGA6 is vital for the proliferation and maintenance of EVI1-high acute myeloid cell lines [21], while another example is the direct interaction and reciprocal regulation between ITGA6 and PSMC2, which interferes with the growth and progression of hepatocellular carcinoma [22]. Elevated levels of ITGA6 inhibit myeloma cell invasion, whereas decreased ITGA6 expression promotes myeloma metastasis [23]. In 10–26% of prostate cancer tumors, cells displayed an invasive appearance and were less adherent, with lower production of the laminin subunit beta-3 precursor (LAMB3) and ITGA6 [24]. Moreover, parthenolide (an antiproliferative drug) influences cellular biological behavior, morphology, and apoptosis induction in human thyroid cancer cells, accompanied by upregulated *itga6* mRNA expression in treated cells [25]. De Archangelis and collaborators categorized integrin α6β4 as a tumor suppressor in the colon of mice models because depletion of the ITGA6 subunit in intestinal epithelial cells causes persistent inflammation leading to tumor growth [26].

## 2. Molecular Mechanisms Regulating the Expression of ITGA6

### 2.1. Transcriptional Regulation of itga6 by Transcription Factor 

It has been proposed that *itga6*, *itga3*, and *itga7* might have a common ancestor [27,28]; however, ITGA6 is encoded by the *itga6* gene located in chromosome 2 (2q31.1) [29], and its expression is regulated at multiple layers. Multiple transcription factors, including MYC, RUNX1, HIFs, and FOSL1, bind to the *itga6* promoter region and enhance its transcription [8,28,30,31,32] (Figure 1A). On the other hand, the nuclear factor of activated T cells (NFAT1) and KLF9 suppress its transcription [8,33] (Figure 1A).

### 2.2. Alternative Splicing Regulation of itga6

The pre-mRNA of *itga6* undergoes alternative splicing to form two splicing variants named *itga6 A* (ITGA6-A) and *itga6B* (ITGA6-B) [28,31]. This alternative splicing of *itga6* is regulated by epithelial splicing regulatory proteins 1 and 2 (ESRP1 and ESRP2), RNA binding motif protein 47 (RBM47), RNA-minding protein muscle blind (MBNL1), RNA-binding protein FOX2 homolog (RBFOX2), and polypyrimidine tract-binding protein 1 (PTBP1) [28,31,34] (Figure 1B). The ITGA6-A and ITGA6-B splice variants of ITGA6 have different functions and expression profiles in different tissue types [11]. The C-terminus of membrane-anchored proteins, such as integrins, is recognized by the PDZ domain, a structural fold found in many signal molecules [7]. In the final three amino acids of the cytoplasmic tail of ITGA6-B, the PDZ-binding motif changes from the normal SDA sequence in ITGA6-A to an alternative atypical SYS motif [35] (Figure 1B). Different binding affinities of downstream effector proteins are influenced by these various sequences [7,34]. Although both ITGA6-A and ITGA6-B isoforms are expressed in CD34+ and CD34+CD38− marrow stem/progenitor cells [7], their signaling pathways differ. The protein kinase C-dependent activation of MAP kinases and mobilization of hematopoietic stem/progenitor cells in the bone marrow stem cell niche during hematopoiesis are caused by ITGA6-A/ITGB1, but not ITGA6-B/ITGB1 [7]. Human embryonic stem cells also expressed both spliced isoforms of ITGA6; however, ITGA6-A is required to inhibit the kinase activity of the focal adhesion kinase (FAK) and prevent differentiation [36,37], while in breast cancer stem cells, the ITGA6-B isoform is responsible for maintaining the stemness [38].

A tumor-specific version with a smaller molecular weight (70 kDa) of ITGA6, α6p, has been identified in various human cancer cell lines. Rather than alternative splicing of precursor *itga6* mRNA, the proteolytic removal of the laminin-binding domain from the tumor cell surface by urokinase-type plasminogen activator (uPA; a pro-metastatic protein) creates α6p [7,18]. Inhibiting the ITGA6 cleavage into α6p drastically slows down tumor development and migration and improves overall survival [7,18] (Figure 1C).

Although ITGA6 is implicated in the invasion and metastasis of a variety of malignancies [33], its biological effects appear to be tissue-specific [39]. The ITGA6-B isoform (mesenchymal cell morphology), rather than the ITGA6-A isoform (epithelial cell shape), identifies breast cancer stem cell populations more precisely and has a stronger potential to produce mammospheres [7,34]. ESRP-1 has been discovered as a critical regulator of the ITGA6-A isoform but a repressor of ITGA6-B [7,34] [40], while MBNL1, a splicing regulator that promotes the production of the ITGA6-A isoform and inhibits metastasis, is suppressed [34] (Figure 1B).

Unlike breast cancer stem cells, proliferative undifferentiated human colorectal cancer cells typically express an ITGA6-A isoform that activates the Wnt/β-catenin pathway leading to tumor growth [7,11,28]. Accordingly, ESPR2 is the main splicing factor responsible for ITGA6-A expression in colorectal cancer cells. Both ITG6-A and ESPR2 expressions are controlled by MYC, which is a target gene of Wnt/β-catenin, all of which suggest a feed-forward loop for ITGA6-A isoform. However, ITGA6-B expression is mostly seen in quiescent differentiated colorectal cancer cells, and its experimental overexpression results in the inhibition of MYC activity [28].

### 2.3. Regulation of itga6 by MicroRNAs

*itga6* mRNA is also susceptible to microRNA (miR) regulation. It has been reported that miR-143-3p, a tumor suppressor, reduces tumors by targeting and degrading *itga6* [41,42] (Figure 2A). In patients with salivary adenoid cystic carcinoma, ADAMTS9-AS2, an antisense transcript of the protein-coding gene ADAMTS9, is highly elevated, competitively binds to miR-143-3p, and hinders *itga6* miRNA-mediated degradation [41] (Figure 2B). In hepatocellular carcinoma, a positive connection between YTHDF3 and *itga6* expression was reported. miR-448 targets and degrades *ythdf3* and *itga6*, which prevents cells from self-renewing [42] (Figure 2C).

It has been shown that miR-302b overexpression in combination with cisplatin is able to impair tumor growth in TNBC-xenografted mice. This effect is mediated by an indirect silencing of *itga6*, as miR-302b targets the transcription factors E2F1 and YY1, which directly regulate *itga6* mRNA expression [43]. *Itga6* is also regulated by miRNA-92a, as shown in ovarian cancer cell lines, repressing their proliferation and metastasis. On the other hand, the long noncoding RNA OIP5-AS1 prevents the repressive action of miRNA-92a on *itga6* in ovarian cancer [44]. Finally, miR-30e-5b also regulates the expression of *itga6*, as shown in colorectal cancer and non-small lung cancer development. In two cases, the repression of itga6 by miR-30e-5b resulted in a reduction in cell proliferation, migration, and invasion [45,46].

### 2.4. N6-Methyladenosine Modification of itga6 mRNA

Multiple RNA-related functions, such as RNA stability, alternative splicing, and translation, are regulated by N6-methyladenosine (m6A) modification. The m6A methyltransferase complex (“writer”), which consists of methyltransferase-like 3 and 14 (METTL3 and METTL14), catalyzes m6A modifications, which can be reversed by m6A demethylases (“erasers”), such as the fat mass- and obesity-associated protein (FTO) and ALKBH5. The YTHDF2 protein, YT521-B homology (YTH) domain-containing protein, helps m6A-modified transcripts degrade faster, while YTHDF1 and YTHDF3 help m6A-modified target RNA translation more efficiently. Numerous m6A-modified target genes play critical regulatory roles in tumor incidence and progression; in addition, the m6A modification might serve an oncogenic or tumor-suppressive function in diverse cancers [8].

In bladder cancer cells, the m6A writer METTL3 and the eraser ALKBH5 changed cell adherence by modulating *itga6* mRNA expression. Although *itga6* mRNA stability and the rate of degradation of the ITGA6 protein remained unaffected, the fraction of *itga6* mRNAs undergoing active translation dramatically increased due to raising polysome-bound *itga6*-mRNA levels [8]. On the other hand, METTL3 promotes the translation of *itga6* mRNA by upregulating the m6A modification of *itga6* in the 3′UTR region [47], which induces the binding of YTHDF1/YTHDF3 to *itga6* mRNA, boosting its protein translation and helping with bladder cancer development and relapse [8,42].

As the first RNA demethylase, the FTO modulates the stability of cellular mRNA by eliminating m6A residues. Many kinds of cancers, including human cervical squamous cell carcinoma tissue and stomach cancer, have high levels of FTO, while bladder cancer has considerably lower expression of FTO levels compared to neighboring normal tissues. The m6A levels increase as the expression of FTO is lowered in a dismal prognosis of bladder cancer patients. The overexpression of FTO in human urinary bladder cancer cell lines (HT-1197 and HT-1376) drastically reduces cell proliferation and invasion capacities while considerably increasing cell apoptosis. Alterations in m6A methylation levels mostly emerge in the 3’UTR of *itga6* transcripts in response to FTO overexpression. Accordingly, after FTO overexpression, *itga6* mRNA and protein are upregulated in bladder cancer cells [48].

Histone demethylase lysine-specific demethylase 5B (KDM5B) has been identified as an overexpressed oncogene in a number of malignancies. In the malignant phenotype of hepatocellular carcinoma, KDM5B inhibited miR-448 expression, which, as mentioned earlier, targets and degrades YTHDF3 and *itga6*. Thus, KDM5B promotes the self-renewal of these cells [42] (Figure 2D).

## 3. Overview of Integrin Signaling

Integrins can both directly and indirectly—through other receptors—stimulate intracellular signaling. Integrins form complexes with receptor tyrosine kinase (RTK), which subsequently interferes with RTK activation by regular ligands. Integrins, for example, directly bind to integrin-linked kinase (ILK), which forms multiprotein complexes with a number of critical components involved in cytoskeletal dynamics and signaling pathways, causing Akt to be phosphorylated. ILK/Akt is a signaling pathway important for leukemic cell survival [12,34,49,50,51].

The Src family kinases (SFKs) recruit the focal adhesion kinase (FAK) through integrin beta subunits [52,53], while the integrin alpha subunits can also activate ERK/MAPK via SFK in an FAK-independent way [53]. Through the transmembrane protein caveolin-1, the α-integrin subset interacts with SFKs such as Fyn and the Shc-adaptor protein. Then, the recruitment of growth factor receptor-bound 2/son of sevenless (GRB2/SOS) and the activation of ERK/MAPK signaling downstream of Ras occurs [7,10,50,53,54,55,56].

### 3.1. ITGA6 Integrin Signaling

The association of integrin α6β4 with laminin substrates results in the activation of Rac1, PKC, PI3K, and ERK signaling pathways, which has been associated with increased carcinogenesis and tumor cell survival [57]. The signal transduction downstream of ITGA6 has received little attention [15]; however, it is known that ITGA6 boosts the activation of PI3K/Akt and MEK/Erk, two key signaling pathways involved in tumor development, which leads to decreasing p53 levels and the promotion of angiogenesis, metastasis, and invasion [41,58]. In leukemia, the enzyme PI3K plays a critical role in transducing extracellular signals that control cell proliferation, survival, and migration [59].

The ablation of ITGA6 from Ph+ acute lymphocytic leukemia cells results in cell cycle suppression and an increase in the proportion of cells in the G0/G1 and G2/M phases [15,59]. The long-term loss or ablation of ITGA6 has been related to caspase activation and increment in apoptosis over time that is associated with an increase in cleaved poly (ADP-ribose) polymerase and p53 expression [15,60], whereas cleaved caspase levels are not modified in P5G10 (ITGA6 antibody) therapy [15].

After the ablation of the *itga*6 in BCR-ABL (Ph+) acute lymphocytic leukemia cells, the phosphorylation of the docking protein CASL increases, while the phosphorylation of SFK substrates decreases. The phosphorylation of specific tyrosine (Y) in Lyn and Fyn has also been shown in this situation, which decreases their tyrosine kinase activity [15]. However, FAK, Fyn, paxillin, and Src proteins involved with FAK signaling and parts of focal adhesions do not exhibit any detectable phosphorylation in protein lysates from undifferentiated hESCs and hiPSC lines with mainly ITGA6 expression [36]. In addition, although Src kinase inhibitors cause growth arrest and death in chronic myeloid leukemia Ph+ cell lines, the same treatment merely reduces the growth of initial Ph+ B-acute lymphocytic leukemia cells and has no effect on their survival. Thus, in B-acute lymphocytic leukemia cells, the signal transduction mechanisms and intracellular protein complexes following ITGA6 binding are only slightly reliant on SFK members. Nonetheless, Src inhibitors diminish the production of ITGA6 in B-acute lymphocytic leukemia cells [15]. Noteworthy, in pancreatic and melanoma cancers, active KRas and BRAF stimulate the production of the pro-tumorigenic ITGA6 through the ERK pathway [56].

### 3.2. Crosstalk between ITGA6 and Other Signaling Pathways in Cancer

In cancers, integrins interact with growth factors, growth factor receptors, cytokines, oncogenes, and enzymes [50,53,54,56,61,62] and then activate or amplify proliferation, invasion, and survival signaling pathways (e.g., MAPK/ERK, PI3K/AKT, Shc, and Rac) [54,56]. Integrin α6β4 combines with the EGFR, ERBB2, and c-Met RTKs to amplify oncogenic signaling in several carcinomas [56]. Activation of signal transducer and activator of transcription-3 (STAT3) and Jun by α6β4-ERBB2 (EGFR) causes cell polarity loss and hyperproliferation, respectively [63]. In glioblastoma stem cells, ITGA6 and fibroblast growth factor receptor 1 (FGFR1) have a synergistic effect in their development and tumor spheroid growth [64]. Integrins also interact with growth factors by binding directly to them—for example, with insulin-like growth factor 1 (IGF1)—and then forming a ternary complex with the corresponding receptor (integrin/IGF1/IGF1 receptor (IGF1R)). The binding of IGFs with integrin α6β4 caused CHO cell (IGF1R+) proliferation. Growth factor mutants are unable to attach to integrins; however, they still bind to their own receptors, operate as antagonists, and consequently suppress angiogenesis, usually caused by wild-type ones [61,65]. This strategy can be helpful in the treatment of cancers with integrin α6β4 overexpression [61].

## 4. ITGA6 as Biomarker of Cancer Stem Cells

Depending on the kind of cancer, cancer stem cells are estimated to make up about 0.05–41% of all tumor cells [39,40]. Self-renewal, asymmetric division, and stimulation of epithelial-mesenchymal transitioning (EMT) are features of cancer stem cells [66,67]. These cells, also known as tumor-initiating cells (TICs), are responsible for cancer metastasis, resistance, and recurrence even after the majority of tumor cells have been removed [39,47,66,67,68] (Figure 3).

ITGA6 was initially described as a stem cell marker in keratinocyte stem cells with the phenotype CD49f-bright/10G7-dim in 1998 [69]. Since then, more than 30 adult stem cells and cancer stem cells have been identified as displaying significant quantities of ITGA6 [37,50,70]. ITGA6 was first discovered as a cancer stem cell or leukemic stem cell marker in acute myeloid leukemia and has been linked to its resistance [15]. Importantly, the expression of ITGA6 is conserved across numerous mammalian species in several of these populations [37]. ITGA6 has been involved as a key regulator of self-renewal, proliferation, spheroid (anchorage-independent growing ability), and tumor formation capability in cancer stem cells [58,70,71,72]. ITGA6 is known to be a better marker than CD133 and CD44 in sphere colony-forming cell cultures of prostate cancer cells [71]. In immune-compromised mice, knocking down *itga6* with shRNA prevents tumor sphere formation and greatly lowers tumor development. The tumorigenicity of breast cancers decreased after siRNA targeting of *itga6* [38]. ITGA6 has been identified as a glioblastoma stem-like cell marker. The development of glioblastoma cancer stem cells has been inhibited by shRNA targeting exons 2 and 14 of *itga6* [73]. In addition, repeated limit dilution transplantation tests of squamous cell carcinomas indicated that integrin α6β1-high populations generate additional cancers, while integrin α6β1-low populations cannot, regardless of whether the cells were CD34-low or CD34-high [50]. Similarly, ITGA6 alone has been used to isolate highly tumorigenic colon cancer stem cells. Furthermore, in the absence of ITGA6, carcinogenesis is not found in CD166-, CD44-, or CD133-positive cells [40,74,75]. Telomerase reverse transcriptase (TERT) has a well-known role in carcinogenesis and, together with ITGA6, is recognized as a stem cell marker in breast cancer cell lines and has been described as having a substantial direct association with malignant tissue [76].

### 4.1. ITGA6-Driven Signals of Stemness

The self-renewal of human embryonic stem cells (ESCs) and human induced pluripotent stem cells (iPSC, reprogrammed from differentiated cells) mainly relies on integrin α6β1 signaling pathways [36]. These pluripotent stem cells express both the ITGA6-A and ITGA6-B isoforms, and the ITGA6-A is able to block the activation of ITGB1, which prevents the phosphorylation of FAK and, consequently, differentiation [36,37]. In contrast, during differentiation, the ITGA6 level diminishes, FAK is phosphorylated (activated), and the expression of pluripotent transcription factors is reduced [36]. During directed differentiation of hESCs, the ITGA6-A isoform is downregulated at a faster rate than the ITGA6-B isoform [36]. Accordingly, ITGA6-B expression is more ubiquitous than ITGA6-A expression, and it is seen at all stages of embryo development [7]. Noteworthy, ITGA6 is important for maintaining stemness in pro-neural glioblastoma stem-like cells but not in mesenchymal glioblastoma stem-like cells [73].

KLF9 suppresses both glioblastoma cell stemness and tumorigenicity by binding to the *itga6*’s promoter region and repressing its transcription [77], whereas Y-box-binding protein-1 (YB-1), an oncogenic transcription/translation factor, induces the expression of *itga6* mRNA, leading to enhanced self-renewal, mammosphere growth, and resistance to paclitaxel treatment [54]. Similarly, the pluripotent transcription factors OCT4 and SOX2 bind to the promoters of *itga6* and enhance its transcription [78] (Figure 3).

Overexpression of *itga6*, an HPV (human papillomavirus) receptor with carcinogenic potential, enhances stemness, drug resistance, and EMT phenotypes in HPV+ head and neck squamous cell carcinoma cells, partially via the PI3K/AKT pathway [58].

### 4.2. ITGA6 and Epithelial-Mesenchymal Transition (EMT)

EMT is a complicated process that organizes particular changes in cellular phenotype, and it is often accompanied by cell adhesion loss and cell polarity [79]. EMT causes epithelial cells to acquire stemness, allowing them to convert into mesenchymal-like cells with improved motility, invasion, metastasis, and death resistance [47,66,70,80]. Autocrine stimulation of the transforming growth factor (TGF)-β signaling pathway leads to increased expression of EMT transcription factors such as twist-related protein 1 (TWIST1), zinc finger E-box-binding homeobox 1 (ZEB1), and Snail2 [47,70,81] (Figure 4A). TGF-β1 has also been linked to cancer cell invasion and to EMT in a variety of human cancers, and it is known to be regulated by ITGA6 [82]. In colorectal cancer cells, it was demonstrated that overexpression of itga6 increased TGF-β1 signaling, while the opposite was observed by siRNA targeting of *itga6* and by overexpression of miR-3940-5p, which targets *itga6* [82]. In addition to these transcriptionally regulated changes, EMT is also regulated post-transcriptionally via alternative splicing regulatory proteins 1 and 2 (ESRP1 and ESRP2) or by muscle blind-like splicing regulator 1 (MBNL1) [11] (Figure 4B,C).

### 4.3. Changes in Matrix Composition/Structure and ITGA6 Expression

The bulk of stem cell biomarkers are proteins found at the cell membrane with the potential to adhere to matrix molecules or other cells to maintain self-renewal [78]. Laminin deposition in stem cell niches might be caused by both non-stem cells and stem cells. Integrin signaling via ILK may control the deposition of laminins by stem cells in the niche of CD34+/CD49f-high hair follicle stem cells [83]. Human embryonic stem cells remodel the extracellular microenvironment by depositing a laminin-511-rich substrate [36,84] rich in laminin-α5 (LAMA5), the primary ligand of integrin α6β1 [36,84,85]. This remodeling of the extracellular matrix is necessary for the survival and expansion of the cells. The knockdown of *LAMA5* resulted in a reduction in ITGA6 expression and differentiation [36,84]. However, in the development of mouse tails, LM532-ITGA6 interactions cause the development of ventral ectodermal ridge progenitor cells [86]. Finally, although ITGA6 is more selective for laminin-111 binding than fibronectin, low-density mesenchymal stem cells treated with TGF-β enhance *itga6* expression only on fibronectin substrates, indicating the confluence effect [78].

Cancer stem cells (CSCs) are closely linked to tumor microenvironments (TMEs) or niches that provide positive crosstalk signals [58], supporting quiescence, self-renewal, and differentiation into progenitors or terminally differentiated cells, which migrate from the niche to perform specialized functions (reviewed by authors of [37,40]). Specific laminins are found in the TME and aid self-renewal and quiescence of CSCs, whereas they increase proliferation, migration, vasculogenesis, and metastasis in differentiated tumorigenic cells [40]. Mesenchymal-like breast cancer stem cells deposit laminin-511, a ligand for integrin α6Bβ1, and activate the Hippo transducer TAZ, promoting cancer cell self-renewal and tumor initiation [87]. ITGA6 and adhesion to laminin-332 are upregulated when the nuclear transcription factor ecotropic viral integration site-1 (EVI1) is overexpressed. The enhanced adhesion by ITGA6 in EVI1-high acute myeloid cells makes them quiescent and promotes long-term self-renewal [21].

## 5. Involvement of ITGA6 in Cancer Metastasis

Rather than tumor formation at its originating location, tumor cell metastasis is responsible for the majority of cancer-related deaths [88]. The following are the major events occurring when a tumor spreads: (i) an increment in tumor mass leading to hypoxia-induced tumor angiogenesis; (ii) remodeling of the ECM for premetastatic niche formation; (iii) the loosening in adherence of tumor cells/EMT; (iv) invasion through the basal membrane that supports the endothelium of local blood vessels; (v) the intravasation of tumorigenic cells into blood and/or lymphatic vessels, and the survival in circulation; (vi) the adherence of circulating tumor cells to the endothelial cell lining of the target organ site; (vii) the extravasation of tumor cells; and (viii) the eventual colonization and growth of secondary tumors at the invaded anatomical/organ site [53].

Integrins have been found to play a vital role in numerous aspects of tumor metastasis [12,89]. In particular, ITGA6 and Thy1 overexpression has been linked to angiogenesis and/or metastasis [82,88,90]. Nevertheless, the involvement of integrins in various types of tumors is varied and complex [12,91]. For example, integrin α6β4 has been introduced as an inhibitor of cancer metastasis in some studies [12,92], while ITGA6 antibodies do not prevent the migration phenotype of melanoma cell lines [49]. The following subsections are intended to highlight the involvement of ITGA6 in multiple processes that support metastasis.

### 5.1. ITGA6 Role in Hypoxia

A tumor is a complex biomass encompassing diverse cancer cells, the tumor microenvironment, stroma, and immune cells infiltrating the tumor [93]. When the tumor is still relatively small and undetectable in the body, the major route of transportation for necessary growth agents, cell nutrition, and waste exports is provided by diffusion rather than circulation. Tumors devote a great part of their energy consumption to growth; therefore, a downregulation of metabolic demand and tissue acidification define the hypoxic parts of a tumor, which is a physiological result of rapid growth and insufficient circulation [67]. Stimulated by hypoxia, the hypoxia-inducible factor (HIF) can activate transcription factors that target genes involved in tumor microenvironment remodeling, tumorigenesis, and metastasis processes such as metabolism, growth, EMT, angiogenesis, immune evasion, and resistance [30,53,67,90,93,94,95].

Otto Warburg was the first to describe tumor cells’ abnormal metabolic function [96,97]. Recent research has found that cancer cell adhesion, which is a key regulator of leukemia growth and chemoresistance, may either induce or be induced by metabolic reprogramming-related cell signaling pathways [95]. ROS may activate numerous oncogenic pathways (e.g., PI3K/AKT, NF-κB, and p38/MAPK) by stabilizing HIF1 [98]. Noteworthy, the ROS-dependent expression level of HIF-1α is elevated in EVI1-high acute myeloid leukemia-transformed cells with ITGA6 overexpressed [99]. In the *itga6* promoter, three potential hypoxia response elements (HREs) have been described for the HIF transcription factors [100]. Furthermore, cells that produce high amounts of ITGA6 are more likely to express HIF1α and HIF-dependent target genes. The expression of *itga6* is reduced at the mRNA and protein levels in HIF-α knockout (KO) of PyMT and MDA-MB-231 cell lines. As a result, selecting cells with high levels of ITGA6 may also result in the selection of cells with a better hypoxic response. HIF signaling also controls the invasion of metastatic breast cancer cells via upregulating ITGA6 [30,67]. Therefore, integrin α6β1 is required for survival under hypoxic stress and for metastatic potential [30]. HIF also selectively enhances the expression of approximately 100 tumor-associated genes, including the proto-oncogene *c-Myc* and genes involved in the phenotype of cancer stem cells [30,67,101]. The direct binding of HIF-2 to the HRE in the promoter regions of stemness transcription factors (Sox2, Nanog, and Oct4) upregulates *itga6* expression [67].

The excess synthesis of ECM proteins (e.g., collagen, laminin, and fibronectin) characterizes the remodeled matrix [40,102]. Upregulation of HIF catalyzes ECM remodeling through restructured adhesion with different components [40,56]. ITGA6 overexpression was discovered in response to the stiffening of a polyacrylamide matrix. Moreover, partially through integrin α6β4, Rac1, and PI3K signaling pathways, increased ECM stiffness induces malignant phenotypes in normal epithelial cells [30]. Vasculature seeps into the hypoxic tumor mass in the next phase when TME cells release proangiogenic substances, such as different matrix metalloproteinases (MMPs), growth factors, cytokines, TGF-β, and vascular endothelial growth factor (VEGF) [40,56]. Integrins regulate cell migration and invasion by affecting the activity of matrix-degrading proteases such as uPA and MMPs [50,103]. Furthermore, α6β4 controlled laminin-511 secretion in brain endothelial cells to increase arteriolar remodeling and vascular integrity in response to hypoxia [104].

### 5.2. ITGA6 and Angiogenesis

The role of ITGA6 in promoting angiogenesis is well documented. For instance, ITGA6 and laminin control the expression of angiogenesis genes such as *cxcr4*, a proangiogenic chemokine receptor, and *angpt2* [104]. Furthermore, the YAP/TAZ transcriptional coactivators, which are candidates for the control of *cxcr4* and *angpt2* during endothelium morphogenesis, are known to be activated by laminins and integrins. ITGA6 is necessary for the deposition of laminin-511 and the stability of endothelial tubes once tubes have been formed [104]. In addition, integrin α6β4 is important in signaling the initiation of angiogenesis during cutaneous wound healing, although its expression is downregulated in newly formed vasculature [104].

ITGA6 promotes tumor angiogenesis. The overexpression of *itga6* significantly enhances lymphangiogenesis and lymphatic metastasis in vivo, playing an important role in metastasis to the lymph node and poor prognosis of lung adenocarcinoma patients [105]. Despite the high level of vascularization, antiangiogenic medicines have failed to achieve the desired results [106,107,108]. The role of circulating angiogenic cells in blood vessel creation is often neglected as a crucial component of neovascularization. Circulating angiogenic cells travel in the circulation to the target tissue via chemotaxis, bind to endothelial cells via integrins, and penetrate the tissue by producing proteinases (MMPs). Glioma-circulating angiogenic cells have a gene expression profile with greater tumor homing capacity (*cxcr4*, *itga6*) and more robust proangiogenic potential compared to circulating angiogenic cells from the umbilical cord (developmental neovascularization) and healthy individuals [109]. Circulating angiogenic cells are also reprogrammed while in circulation by the target tissues to meet their needs [107].

### 5.3. ITGA6 in Extracellular Vesicles

Extracellular vesicles (EVs), or exosomes, are nano-sized particles that are produced by a variety of cells and have the ability to convey specific payloads such as nucleic acids, microRNAs (miRs), proteins, and lipids; they have been identified as a critical mediator in cell-to-cell communication [66]. Cancer cells can be “educated” by exosomes to reach a certain tissue. Exosomal organotropism is further governed by integrins such as ITGA6, which facilitated exosome absorption by particular tissue-resident stromal cells within target tissues [8,12,56,66,102,110].

Importantly, certain exosomal integrins found in cancer patients’ plasma samples corresponded with the location of metastases, suggesting that they might potentially predict future metastatic locations at the time of diagnosis [8,12,56,66,102,110]. The determination of ITGA6-A isoform expression in blood EVs in pancreatic ductal adenocarcinoma patients could be a useful blood marker for its early recurrence diagnosis, whereas the ITG6A-B isoform is predominantly expressed in EVs from the blood of normal volunteers. In EVs, high expression of ITGA6-A isoforms occurs significantly sooner than CA19-9 [111]. Furthermore, following surgery, the expression of ITGA6-A isoforms in EVs is reduced dramatically but then increases several months before clinical recurrence [111].

Through targeting *ITGA6*, mesenchymal stem cell-derived exosomal-miR-3940-5p suppresses colorectal cancer cell invasion and EMT [82,112,113]. Similarly, by targeting ITGA6 and following the TGF-β1/Smad pathway, EV-miR-127-3p reduced EMT in highly aggressive choriocarcinoma cells. Thus, miR-delivering EVs have emerged as a unique and potential therapeutic strategy for tumors and metastasis [114].

### 5.4. ITGA6 and Metastasis to the Central Nervous System (CNS) of Acute Lymphoblastic Leukemia (ALL) Patients

Because most chemotherapeutic medications are unable to cross the blood-brain barrier (BBB), leukemic cells buried in the CNS are unable to be efficiently removed and hence become the source of recurrent extramedullary leukemia [94,115,116]. Blocking CXCR4, the receptor for the chemoattractant CXCL12, only partially inhibits acute lymphoblastic leukemia cell invasion of human cerebrospinal fluid. Furthermore, in vivo, continuous CXCR4 inhibition does not prevent acute lymphoblastic leukemia xenografts from developing CNS invasion [59]. Acute lymphoblastic leukemia cells’ motility is modulated by PI3K independent of AKT and ROCK; however, it is strongly reliant on the FAK signaling pathway [59]. Aside from motility, PI3K inhibition reduces cell surface expression of ITGA6 in acute lymphoblastic leukemia cells and limits CNS metastasis [56,59,117]. ITGA6 is strongly expressed in most acute lymphoblastic leukemia patients, and its interaction with laminin enhances the development of CNS disease [5,59,94,118,119] and the maintenance of minimal residual disease (MRD) after chemotherapy [15].

For a long time, it was assumed that leukemia cells entered the CNS through the bloodstream [117]. However, it has been demonstrated that acute lymphoblastic leukemia cells travel straight from the bone marrow to the CNS surrounds via the laminin-rich ECM of emissary bridging arteries rather than metastasizing within the circulation as in the case of solid tumor cells [59,73,95,110]. Moreover, the dural lymphatic system provides a new lymphocyte trafficking channel, and acute lymphoblastic leukemia cells use CNS lymphatics to enter or exit the CNS [117]. The expression of ITGA6 also mediates the invasion of CNS tissues via neuronal migration pathways [59,94].

Specific integrin binding was predicted to have a role in CSF and CNS invasion and to be highly expressed by acute lymphoblastic leukemia cells in diagnostic bone marrow samples [120]. ITGA6 expression in acute lymphoblastic leukemia cells from archival bone marrow biopsies of individuals with acute lymphoblastic leukemia is related to the incidence of CNS relapse [59], irrespective of other criteria thought to enhance the probability of CNS disease involvement [59,118]. In contrast, another study claimed that ITGA6 positivity in pediatric B-acute lymphoblastic leukemia is not associated with an increased frequency of CNS involvement at diagnosis, day 29 MRD, or during relapse [118]. Moreover, *itga6* is shown to be downregulated in Nalm-6 cells isolated from CNS compared to bone marrow [121]. According to Scharff and collaborators, BCP-acute lymphoblastic leukemia distribution to the CSF is adversely linked with ITGA6 [120]. Furthermore, rather than ITGA6 expression, subsequent investigations found that ITGA5 and ITGA9 expressions are positively linked with CSF colonization in primary B-acute lymphoblastic leukemia samples [95].

CNS relapses are reported to be common in patients with t(9;22) BCR-ABL1 (Ph+), *MLL* rearrangement (*KMT2A*), hypodiploidy, or TCF3-PBX1 fusion [94]. Ph+ leukemic cells also show an immature B-cell profile with high expression of ITGA6 [117,122]. In contrast, in spite of common CNS involvement, ITGA6 expression is low in B-acute lymphoblastic leukemia with *KMT2A* rearrangement. Considering no *KMT2A*-rearranged cases in Yao et al.’s study of B-acute lymphoblastic leukemia specimens for evaluation of ITGA6 expression, it is possible that molecules other than this integrin are involved in B-acute lymphoblastic leukemia with *KMT2A* rearrangement infiltrating the CNS [59]. Moreover, Prieto and colleagues proposed that NG2, significantly expressed in *KMT2A*-rearranged B-acute lymphoblastic leukemia, is important in their CNS invasion. Other explanations can also be that rare blasts with higher ITGA6 levels can enter the CNS or that ITGA6 expression is elevated throughout the CNS infiltration process [123].

## 6. ITGA6 as Potential Diagnostic Biomarker

ITGA6 has been shown to be elevated in tumor cells despite being expressed at low or undetectable levels in healthy tissues [15,50,55]. Various datasets such as ONCOMINE, GEPIA, TIMER, HPA, Kaplan–Meier Plotter, GEO, and TCGA have been used to examine diverse expression patterns and prognostic values based on ITGA6 in patients with various kinds of malignancies. In addition, immunohistochemical labeling from the Human Protein Atlas database revealed consistent ITGA6 protein levels in cancer cells. Consequently, ITGA6 has been shown to be considerably overexpressed in a variety of tumors, as well as liquid malignancies. ITGA6 expression is linked to tumor development, aggressiveness, increased risk of recurrence, poor patient prognosis, shorter event-free survival, and overall survival time in cancer patients [8,22,31,40,50,57,61,70,103,124,125,126,127,128,129]. Therefore, ITGA6 can be proposed as a prognostic biomarker and therapeutic target in early-stage cancer diagnosis [6,50,58,127,130].

However, various types of malignancies have varied integrin expression patterns, and even contradicting evidence for the same integrin can be discovered within the same cancer type [120]. The mRNA levels of *itga6* in lung adenocarcinoma and normal tissues acquired by The Cancer Genome Atlas and single-cell RNA-sequencing do not differ significantly [129]. Furthermore, ITGA6 is considered a good predictive marker in several cancers, such as stomach adenocarcinoma [131] and multiple myeloma [23]. As a result, better event-free survival and or overall survival are linked to a high level of ITGA6 as an independent prognostic factor. The LncRNA identified as *ITGA6*-AS1 forms an RNA duplex with *itga6* pre-mRNA, increasing the stability of *itga6* pre-mRNA and reducing multiple myeloma cell invasion [23].

In the case of ovarian cancer, there is contradictory information regarding the expression of ITGA6 and its possible effect. Utilizing public datasets, ITGA6 is found to be a good prognosis predictor in ovarian cancer. Decreased *itga6* mRNA expression has been significantly associated with poor progression-free survival (PFS) and overall survival (OS) in ovarian cancer patients [132]. An external dataset validates that ITGA6 is significantly associated with longer OS and better PFS. In this analysis, it was also noticed that higher levels of *itgb4* mRNA have a significant association with poor prognosis, metastasis, and resistance to platinum treatment. However, the heterodimeric partner *itga6* strongly correlates with prognosis and lacks a significant correlation with metastasis and platinum resistance [132]. Givant–Horwitz and collaborators found lower *itga6* mRNA expression in FIGO stage IV ovarian cancer solid tumors compared to stage III ovarian cancer, which was correlated with shorter overall survival [133]. On the other hand, Villegas–Pineda and collaborators reported that blocking ITGA6 decreases migration and invasion of ovarian cancer cells. Moreover, its blocking partially sensitized ovarian cancer cells to carboplatin [134]. ITGA6 expression has been found to be significantly downregulated in high-grade serous ovarian cancers compared to normal counterparts [135]. *ITGA6* adhesion mediates drug resistance in ovarian cancer, indicating that its expression is upregulated in cisplatin-resistant cells [136].

As previously indicated, both Thy1 and ITGA6 are potential markers of various cancer stem cells, and their expression is associated with poor differentiation, large tumor growth, lymph node metastases, high invasiveness, and shorter overall survival [39]. In breast cancer, ITGA6 expression is recognized as a stronger predictor of poor prognosis than other known variables, such as estrogen receptor status [55]. In ER-negative breast cancer, global ITGA6 protein expression is independently predictive of poor outcomes [34]. In acute myeloid leukemia and acute lymphoblastic leukemia cells, ITGA6 overexpression has also been linked to a poor prognosis [15,16,59]. ITGA6 is also substantially higher in cases of acute myeloid leukemia recurrence than in cases of acute myeloid leukemia remission [21]. It has been reported that relapsed B-acute lymphoblastic leukemia and EVI1+ acute myeloid leukemia patients have considerably higher levels of ITGA6 expression [60,137]. Moreover, ITGA6 expression is substantially more common in CD34+ pre-B-acute lymphoblastic leukemia than in CD34- pre-B-acute lymphoblastic leukemia. In addition, the expression of CD49c is also favorably linked with the ITGA6 expression in pre-B-acute lymphoblastic leukemia [138].

Because the majority of CNS relapses occur in CNS-negative children, cytological testing of cerebrospinal fluid (CSF) is the gold standard for diagnosing CNS leukemia; however, its lower sensitivity compared to autopsy reports has also been shown [5,94,95]. The use of imaging to diagnose CNS leukemia lesions might be more convenient and intuitive; however, traditional imaging exams are considered ineffective in CNS leukemia lesion identification. Because CNS leukemia is mediated by ITGA6–laminin interactions, ITGA6 might be a viable target for more sensitive and noninvasive diagnostic approaches to targeted molecular CNS leukemia imaging. The peptide CRWYDANAC (S5), a new ITGA6-targeting peptide, has a stronger selectivity and affinity for ITGA6 than CRWYDDENAC (RWY) [5,130]. In animal models, S5 peptide-based MRI and PET probes demonstrated enhanced ability in imaging CNS leukemia lesions, highlighting their potential for further research and clinical application [5]. By creating a superior MRI signal and a higher sensitivity to tiny hepatocellular carcinoma, which is hardly detected using traditional methods, S5 can provide both functional and anatomical information [130]. In addition, an early phase I clinical study (NCT04289532) employed 99mTc-RWY, a radiotracer for ITGA6-targeting (RWY peptide), to perform single-photon emission computed tomography imaging in breast cancer patients, helping to predict overall survival and prognosis [12].

On the other hand, the ITGA6-B isoform suppresses the cell cycle of colonic epithelial cells and is downregulated in colon cancer, whereas an up-regulation in ITGA6-A expression has been linked to a hyperproliferative phenotype [13,28,139]. Therefore, despite minimal quantities in stools, ITGA6-A might be a promising potential biomarker for a noninvasive stool RNA test in colorectal cancer screening [139,140].

In head and neck malignancies, the expression of *itga6* showed a positive correlation with *itga5* and *itga3* [127,131]. The expression of *itga6* shows a negative correlation with infiltration levels of B-cells and T-CD8+ cells in head and neck cancer cells, while it is favorably correlated with T-CD4+ cells. No significant links have been discovered between ITGA6 and macrophages, neutrophils, or dendritic cell counts [127]. In situations of increased *itga6* expression, patients with HPV+ head and neck squamous cell carcinoma have a poor prognosis; however, *itga6* expression levels have no effect on survival rates of HPV-negative head and neck squamous cell carcinoma patients [58].

## 7. ITGA6 and Cancer Drug Resistance

Drug resistance in tumor cells is developed by adaptive responses to external stimuli, the activation of specific pro-survival signals/antiapoptotic programs, the selection of drug-resistant subpopulations, and changes in microenvironmental properties [12,55]. Integrins play an important role in the development of drug resistance. Laminin-binding integrins are especially important for bone metastatic lesions, as the bone’s laminin-rich environment allows early metastatic cancer cells to survive and resist chemotherapeutic treatments [15,18]. This is due in part to a mechanism known as cell adhesion-mediated drug resistance (CAM-DR) [15,136]. Furthermore, the stiffening of the extracellular matrix makes tumor cells resistant to chemotherapy by reducing antitumor medication penetration [12,30]. Integrins are also overexpressed in tumors, which activate downstream signaling pathways. As a result, cell proliferative signals are triggered without the need for a receptor tyrosine kinase and bypass the inhibiting action of targeted medicines. Therefore, targeting integrins in conjunction with other antitumor medicines (radiotherapy, chemotherapy, and targeted therapy) has the potential to overcome tumor resistance [12,55,141].

According to bioinformatics and experimental findings, *itga6* is a key drug-resistance gene [136]. It suppresses DNA damage and promotes DNA repair [55,73,136]. ITGA6 is involved in the development of poor prognosis of aggressive cancer phenotypes, which can cause extracellular matrix modifications and antiapoptotic mechanisms in order to attain radio/chemoresistance [6,12,21,73,120,136,142,143,144]. Forkhead box protein M1 (FOXM1) and E2F-4 are transcription factors contributing to cell cycle regulation and are most likely influenced by ITGA6 in an ERK-dependent manner [73]. AKT is regarded as a primary mediator of resistance, and ITGA6 plays a crucial role in cancer resistance by modulating the MAPK/ERK and PI3K/AKT survival signaling pathways [6,10,12,21,22,54,55,73,103,120,137,144]. The adjuvant radiation for breast cancer that targets ITGA6 signaling might be a beneficial strategy. PI3K inhibition partially antagonizes the protective effect of ITGA6 against cell death, while ITGA6 depletion inhibits AKT phosphorylation after radiotherapy [55]. PI3K inhibition partially antagonizes the protective effect of ITGA6 against cell death, while *itga6* depletion inhibits Akt phosphorylation after radiotherapy [55]. In addition, the PI3K/AKT pathway may also be involved in the control of drug-carrying pumps [10].

*ITGA6* expression is linked to a group of survival genes, including *birc5*, *mcl1*, and *xiap* [6]. Interestingly, survivin, an antiapoptotic IAP family member, is increased by chemotherapy while reduced by treatment with P5G10, an antibody against ITGA6, in acute lymphoblastic leukemia cells. In addition, the ablation of *itga6* sensitizes acute lymphoblastic leukemia cells to tyrosine kinase inhibitors (TKI) or chemotherapy. Even targeting ITGA6 alone promotes cell death [5]. This also helps leukemic mice live longer and achieves long-term remission [5,15]. In conclusion, this evidence indicates that ITGA6 plays a key role in chemoresistance, relapse, and minimal residual disease [5].

Treatment with the antibody against ITGA6 and P5G10 de-adhered acute lymphoblastic leukemia cells from the stroma and reduced stromal support in the face of chemotherapeutic damage [145]. Additionally, it reduces acute lymphoblastic leukemia cell flexibility as measured by single-beam acoustic tweezers. Resistant and highly invasive cancer cells are more deformable than weakly invasive cancer cells, allowing them to move more easily. Thus, acute lymphoblastic leukemia cells treated with integrin antibodies may become less resistant to chemotherapy [146]. However, the inhibition of ITGA4 has no effect on deformability; hence, ITGA4 and ITGA6 have distinct methods for de-adhesion [147]. The expression of ITGA6 is principally responsible for the increased cell adhesion capability and survival of EVI1-high leukemia cells. Furthermore, in EVI1-low leukemia cells, the overexpression of EVI1 enhances the expression of *itga6*. Finally, drug sensitivity is restored in EVI1-high leukemia cells treated using ITGA6-neutralizing antibodies or short hairpin RNA against *evi1* [21,60,137,142].

### 7.1. ITGA6 and Minimum Residual Disease

The condition under which clinical remission is established and remaining leukemia cells are detected using flow cytometry or polymerase chain reaction (PCR) testing is referred to as minimum residual disease [15]. In precursor B-acute lymphoblastic leukemia, the persistence of minimum residual disease at the completion of induction treatment (day 29) is a well-established prognostic predictor [15,119,120,123,148,149]. Although some mutations are linked to a bad prognosis, their presence at diagnosis does not always indicate relapse: some mutations can be lost at relapse, and others can be found in diagnostic samples that never relapse [68].

Cell adhesion is the most elevated gene network in quiescent/resistant and minimum residual disease, indicating that interactions with the microenvironment are involved in resistant phenotypes [68]. One of the top five elevated genes in CD34-positive leukemia cells is *itga6*, and it is linked to poor therapeutic response [150]. In addition, *itga6* is one of several genes overexpressed in B-acute lymphoblastic leukemia upon diagnosis, and it has been linked to the discovery of minimal residual disease later on [15,119,120,123,148,149]. The expression of ITGA6 may also be upregulated during chemotherapeutic agent treatment [60,123,148], or alternatively, the therapy may select a subset of ITGA6 blasts that are not visible at diagnosis [148].

Evaluation of the genetic subgroups (KMT2A-rearranged (MLL), BCR-ABL1+ (Ph+), ETV6-RUNX1+ (t12,21), hypodiploidy, and hyperdiploid) shows substantial differences in ITGA6 expression, despite the high frequency of its overexpression among cases of B-acute lymphoblastic leukemia in general [123]. In KMT2A-rearranged individuals, the proportion of ITGA6+ blasts as well as mean fluorescent intensity using flow cytometry analysis, is lower than in cells without KMT2A rearrangement, limiting use as a minimum residual disease diagnostic in this genetic subgroup. The proportion of ITGA6+ blasts in hypodiploid and hyperdiploid patients appears to be more varied than in BCR-ABL1+ or ETV6-RUNX1+ instances [123]. ITGA6 expression has been found to be higher among (t12,21) and chromosome 21 amplification (iAMP21) patients [119]. It has also been reported that ITGA6+ is linked to a greater prevalence of ETV6-RUNX1 cytogenetic abnormalities and a lower frequency, although not statistically significant, of KMT2A rearrangement and hypodiploidy, while none of the ITGA6+ patients have iAMP21 or TCF3-PBX1 (t1,19). Patients positive and negative for ITGA6 have similar prevalences of copy number aberrations detected using comparative genomic hybridization of *CDKN2A*, *CDKN2B*, *PAX5*, *IKZF1*, *BTG1*, *ERG*, *TP53*, *RB1*, *EBF1*, and *CRLF2*, indicating that there is no correlation between itga6 and leukemia-related genes [118]. The overexpression of *itga6* at diagnosis is also linked to the existence of minimum residual disease at days 19 and 46 after chemotherapy induction but not linked to a genetic subtype [123,151].

Notably, ITGA6 expression is negatively associated with white blood cell count, as is characteristic of the B-other cytogenetic leukemia category, which cannot be categorized into any of the known cytogenetic groups [119]. In the case of strict bone marrow adherence, lesser amounts of leukemic blasts in circulation may be predicted, resulting in a high minimum residual disease but low white blood cell counts [119,149]. One research study illustrated considerably low ITGA6 surface expression in diagnostic bone marrow samples from individuals with leukemic blasts within cerebrospinal fluid, contradicting the previously postulated role of ITGA6 aiding in leukemic cell propagation to the CNS [119].

All in all, integrin regulation is extremely complicated, and their function is defined by integrin activation and ligand affinity of specific integrin heterodimers rather than the amounts of individual integrin subunits. Therefore, it has been proposed that focusing on integrin affinity states is more likely to uncover the link between integrins and minimum residual disease rather than only focusing on integrin expression patterns [120].

### 7.2. ITGA6 Role in Autophagy

Autophagy is a highly conserved lysosomal degradation mechanism that is important for homeostasis, differentiation, development, and survival [91,152]. Some of the most well-known signaling pathways (e.g., PI3K/AKT/mTOR, p53, JAK/STAT, and RAS) that control crucial cell biology and the cell cycle are all linked to autophagy. In malignancies, autophagy is thought to be a double-edged sword [152]. Misfolded proteins and damaged organelles are removed via autophagy, which helps to avoid carcinogenesis [91,99,152]. Autophagy can also affect the immunogenicity of tumor or stroma cells in the tumor microenvironment and in the establishment of anticancer immunity [91]. On the other hand, autophagy can address the demands of established cancer by supplying materials for biological synthesis through the degradation of damaged organelles and the recycling of macromolecules [91,99,152]. Once faced with a variety of external stimuli, such as famine, hypoxia, or medication, the amplitude of autophagy may rise dramatically in order to deliver nutrition and eliminate toxic compounds [152]. Autophagy has also been linked to treatment resistance in a variety of malignancies [91,99,152].

The increased metabolic activity in cancer cells leads to increased ROS generation since these cells require energy and nutrients to maintain their high rates of cell growth and proliferation. ROS acts as a second messenger, triggering the activation of critical signaling pathways (e.g., PI3K, MAPK, NF-kB, AP-1), as well as the production of HIF-1 for angiogenesis and autophagy genes, all of which are crucial regulators of cell proliferation, metabolism, survival, and cancer development. FLT3-ITD, JAK2, and Ras mutations increase ROS levels in acute myeloid leukemia cells, resulting in genomic instability and leukemogenesis. EVI1, an oncogenic transcription factor, increases survival and treatment resistance by inducing autophagy through elevated intracellular ROS. In leukemia, inhibiting autophagy improves chemosensitivity and tumor cell death [99]. Nonetheless, due to a high quantity of ROS, leukemic cells isolated from the CSF of acute lymphocytic leukemia-bearing animals displayed lower proliferation and viability [95]. Therefore, developing a model based on autophagy-related genes to predict overall patient survival is of critical importance [130,152]. Based on data from The Cancer Genome Atlas, ITGA6 is assumed as an autophagy-related gene, and a Cox analysis indicated that autophagy-related risk scores of ITGA6 is a predictor of lower-grade gliomas and female lung adenocarcinoma patients [91,113]. Ribosomal protein SA (RPSA) has the potential to interfere with autophagy in breast cancer and esophageal tumor cells. Noteworthy, ITGA6 performs its biological tasks by binding to nonintegrin RPSA molecules and facilitates cellular adhesion and function [103,153].

## 8. Potential Therapeutics of Targeting ITGA6

In addition to integrins being involved in crucial aspects of malignant progression, their expression patterns in neoplastic lesions differ from nonneoplastic tissues [12,73]. Therefore, integrins can be valuable probes in cancer prognosis, imaging diagnostic methods, and therapeutic efficacy [5,12]. The overexpression of ITGA6 indicates its potential use as a therapeutic target [15,142] or for the development of novel combination therapeutic strategies to treat refractory cancers [6,124,127]. However, some clinical studies indicate that, in general, the use of integrin-selective inhibitors does not achieve the expected efficacy when used alone or in combinational strategies. Although there are few successful clinical trials, some preclinical studies have encouraging results [12]. There is no evidence that ITGA6-blocking antibody treatment has any deleterious effects on normal tissues in mouse models, whereas the complete ablation of the *itga6* gene has drastic effects [15,18]. In the xenograft human prostate cancer mouse model, treatment with the J8H antibody, which blocks ITGA6 by converting it into ITGA6-p, reduces osteolytic tumor activity and induces a sclerotic reaction in bone lesions [18,154]. The targeting of ITGA6 by the highly specific peptide CRWYDENAC in combination with cisplatin prodrug Pt(IV) suppresses nasopharyngeal carcinoma tumor growth significantly without deleterious side effects in mice [155].

As there are pitfalls to finding an effective and safe therapy capable of inhibiting ITGA6 in vivo, and the functional blocking antibodies available may potentially cause important side effects if systemically administered because of its broad expression [73], another potential strategy to reduce the effects of ITGA6 may be reducing the movement of tumor cells toward pre-existing blood vessels, also called vessel cooption. This therapeutic strategy may be potentially safer than directly targeting ITGA6 [73]. For more information regarding the targeting of ITGA6 for therapy purposes, readers are referred to a recently published review by Zhang and collaborators [156].

## 9. Conclusions

As the roles of integrins in diseases are discovered, we have tried to achieve a further understanding of the mechanisms and signaling specific to individual subunits of different heterodimers. Therapeutic strategies could be made specific by targeting each subunit related to the particular situation. Meanwhile, the potential overlapping or compensating roles of integrin subunits need to be clarified and considered. In this review, we summarized the recent understanding of ITGA6’s function in various cancer types. The role of ITGA6 differs based on several features, such as cell background, cancer type, and post-transcriptional alterations. ITGA6 is overexpressed in numerous types of malignancies, and its expression level is related to cancer development and poor survival in patients. In addition, its expression levels are closely associated with signs of epithelial-mesenchymal transitioning. Exosomal ITGA6 also implies metastatic organotropism. Thus, further understanding of these mechanisms could unravel new therapeutic opportunities that specifically target ITGA6 in certain cells and cancer types, which could lead to better clinical outcomes.

## Figures and Tables

**Figure 1 cancers-15-03466-f001:**
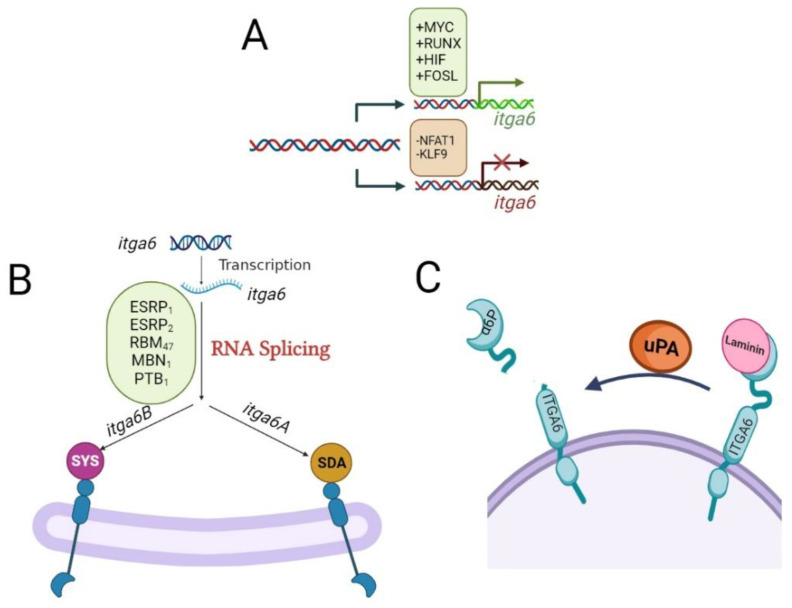
Molecular mechanisms regulating the expression of *itga6*. (**A**) The expression of *itga6* is regulated at multiple layers. Transcription factors, including MYC, RUNX1, HIFs, and FOSL1, bind to the *itga6* promoter region and enhance its transcription, while the nuclear factor of activated T cells (NFAT1) and KLF9 suppress its transcription. (**B**) The pre-mRNA of *itga6* undergoes alternative splicing to form two splicing variants, named *itga6A* (α6A) and *itga6B* (α6B). This alternative splicing of *itga6* is regulated by epithelial splicing regulatory proteins 1 and 2 (ESRP1 and ESRP2), RNA binding motif protein 47 (RBM47), RNA-minding protein muscle blind (MBNL1), RNA-binding protein FOX2 homolog (RBFOX2), and polypyrimidine tract-binding protein 1 (PTBP1). The C-terminus of membrane-anchored proteins, such as integrins, is recognized by the PDZ domain, a structural fold found in many signal molecules. In the final three amino acids of the cytoplasmic tail of ITGA6-B, the PDZ-binding motif changes from the normal SDA sequence in ITGA6-A to an alternative atypical SYS motif. Different binding affinities of downstream effector proteins are influenced by these various sequences. (**C**) The proteolytic removal of the laminin-binding domain from the tumor cell surface by urokinase-type plasminogen activator (uPA; a pro-metastatic protein) facilitates the lighter molecular weight (70 kDa) of ITGA6, i.e., α6p.

**Figure 2 cancers-15-03466-f002:**
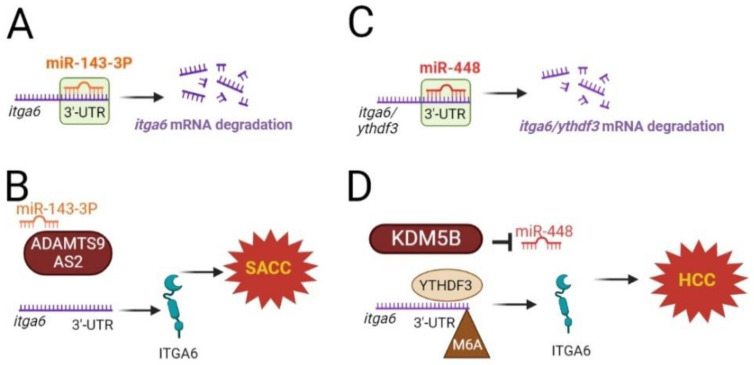
Regulation of itga6 mRNA by microRNAs (miR) and N6-methyladenosine modifications. (**A**) miR-143-3P degrades *itga6* mRNA, and (**B**) in patients with salivary adenoid cystic carcinoma (SACC), the highly elevated levels of ADAMTS9-AS2, competitively bound to miR-143-3P, hinder *itga6* miRNA-mediated degradation. (**C**) miR-448 targets and degrades *ythdf3* and *itga6*. (**D**) On the other hand, the histone demethylase lysine-specific demethylase 5B (KDM5B), an overexpressed oncogene in the malignant phenotype of hepatocellular carcinoma (HCC), inhibits miR-448 expression and therefore protects *itga6* degradation.

**Figure 3 cancers-15-03466-f003:**
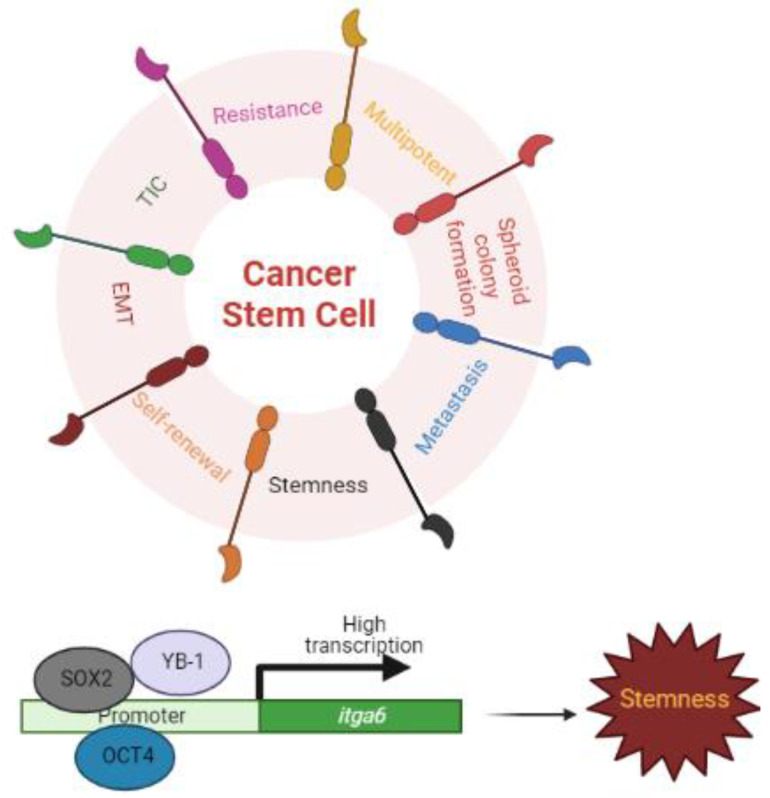
ITGA6 and Stemness. Cancer stem cells (CSCs) display significant quantities of ITGA6, which contributes to all features of CSCs (e.g., self-renewal, asymmetric division, and stimulation of epithelial-mesenchymal transitioning (EMT); tumor-initiating cells (TICs); cancer metastasis, resistance, and recurrence). The Y-box binding protein-1 (YB-1), an oncogenic transcription/translation factor, and the transcription factors OCT4 and SOX2 bind to the promoters of *itga6* and induce the expression of *itga6* mRNA.

**Figure 4 cancers-15-03466-f004:**
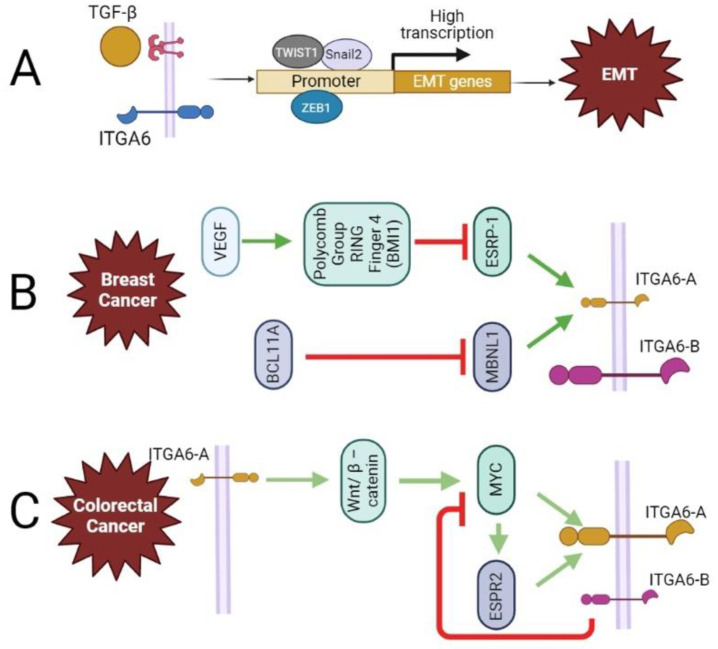
The contribution of ITGA6 in epithelial-mesenchymal transition (EMT) and regulation of its splicing variants. (**A**) In colorectal cancer cells, the overexpression of ITGA6 increases TGF-b1 signaling, which leads to increased expression of EMT transcription factors such as twist-related protein 1 (TWIST1), zinc finger E-box-binding homeobox 1 (ZEB1), and Snail2. (**B**) In triple-negative breast cancer (TNBC), VEGF signaling induces polycomb group ring finger 4 (BMI1)-mediated suppression of ESRP-1, consequently causing a decrease in ITGA6-A and an increase in expression of ITGA6-B (breast CSC population). Moreover, highly expressed in TNBC and involved in stem cell destiny, metastasis, and ECM remodeling, B-cell leukemia/lymphoma 11A (BCL11A) suppresses MBNL1, a splicing regulator that promotes the production of the ITGA6-A. (**C**) Proliferative undifferentiated human intestinal cells (colorectal cancer, CRC) typically express ITGA6-A, which activates the Wnt/β-catenin pathway leading to tumor growth. *MYC* is a target gene of Wnt/β-catenin, and both ITGA6-A and ESPR2 expressions are controlled by MYC. This suggests a feedforward loop for ITGA6-A. However, ITGA6-B experimental overexpression results in the inhibition of MYC activity.

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
