# Peer review of "Regulation and Functions of α6-Integrin (CD49f) in Cancer Biology"

_cancers, 2023, doi:10.3390/cancers15133466_

Round 1

Reviewer 1 Report

The review entitled ' Regulation and Functions of α6-Integrin (CD49f) in cancer biology’. 

The review article is very interesting, detailed, and elaborate. The authors could improve the review article.

1.      English language proof reading of the article is required.

2.      The description about integrins is very minimal in the introductory section. A schematic description of the possible combinations of the integrins is required.

3.      In Section 2, the authors describe about a common ancestor but do not describe what is its name?  

4.      The font size is to small in figure 1.

5.      They need to label ITGA6 in the figure, it does not describe such information in the figure 1 A.

6.      What is the abbreviation of SYS, SDA?

7.      Can the authors refer to the article that shows interaction between Caveolin and ITGA6 and describe what this interaction plays a role in. does it change the conformation of the protein or if there is a complex formation between these proteins. The authors need to elaborate about it.

8.      Fig3, the authors describe about various biological responses activated by this pathway, it would be nice to include them as biological responses or an arrow indicating them.

9.    

Can the authors shed some more light on TGFbeta regulation by ITGA6 as it is unclear in the description in section 4.2?

10.  What are the possible avenues that could be explored to target ITGA6 and in various diseases can be outlined in a separate section.

11.  There are other emerging technologies like mRNA therapy or CRISPR-CAS9, to target such diseases, the authors need to discuss about the recent research findings and the ongoing trials.

12.  The authors can add a section on the ongoing clinical trials and the expected timeline of outcomes.

13.  The discussion is too short, can the authors discuss more broadly in the context of checkpoint inhibitors and their implications in cancer.

14.  The references are in a different font size and the font size on page 15 needs formatting.

15.  The authors described about contradictory studies on page 15 but need to refer to such articles as they are missing.

16. The authors need to describe about the cross talk of ITGA6 with other signaling pathways, even though the authors briefly touched upon it. 

The article needs to be more structured as a lot of information is described randomly under various sections of the review article. 

The references are in a different font size and the font size on page 15 needs formatting.

The font size is to small in figure 1.

Author Response

Reviewer 1 MDPI

  1. English language proof reading of the article is required.

Dear Reviewer, we appreciate your valuable comments and suggestions for our manuscript.  We have performed a proofreading of the manuscript as you requested.  The proofreading was done by us and not by a professional because we did not have enough time to do so.

  1. The description about integrins is very minimal in the introductory section. A schematic description of the possible combinations of the integrins is required.

Thank you for the suggestion.  Accordingly, we added a paragraph in the introduction.  We did not include a schematic description of the possible combinations as requested, due to the limited space that we have, and taking into consideration that our review is already long enough.  However, we briefly described in the text the possible combinations of integrin heterodimers.  This modification can be found on page 2 lines 24-38.

  1. In Section 2, the authors describe about a common ancestor but do not describe what is its name?

A name for the potential common ancestor for the genes itga6, itga3, and itga7 has not been proposed by this hypothesis.

  1. The font size is too small in figure 1.

We have increased the font size in all figures to improve their readability, thanks for pointing this out to us.

  1. They need to label ITGA6 in the figure, it does not describe such information in the figure 1 A.

We labeled correctly the figure and its legend.

  1. What is the abbreviation of SYS, SDA?
  2. Can the authors refer to the article that shows interaction between Caveolin and ITGA6 and describe what this interaction plays a role in. does it change the conformation of the protein or if there is a complex formation between these proteins. The authors need to
    elaborate about it. Fig3, the authors describe about various biological responses activated by this pathway, it would be nice to include them as biological responses or an arrow indicating them.

We want to apologize for the confusion, as it was a mistake from us.  Thus far, the interaction of caveolin and ITGA6 has not been shown, as it has been demonstrated for other alpha subunits.  We removed the figure, as it was not specific for ITGA6 signaling, and to recover space in the manuscript for other topics.

  1. Can the authors shed some more light on TGFbeta regulation by ITGA6 as it is unclear in the description in section 4.2?

The following sentence was added to the manuscript to address this request:

“In colorectal cancer cells it was demonstrated that overexpression of itga6 increased TGF-β1 protein translation, while the opposite was observed by siRNA targeting of itga6 and by overexpression of the miR-3940-5p, which targets itga6”.

This addition can be found in page 11 lines 10-13.

  1. What are the possible avenues that could be explored to target ITGA6 and in various diseases can be outlined in a separate section.

We express gratitude to the reviewer for the appreciation that ITGA6 can be used as a target for therapeutic use.  In our original submission, we briefly touched on this topic in section 8 page 22 lines 3-31, and due to the fact that our review is dedicated to understanding the role of ITGA6 in cancer biology and not therapeutics.  However, we follow the recommendation and added further information (page 22 lines 1-6), and in addition, we recommended readers look for a recently published review on this topic.  We added the following to this section:  “For more information regarding the targeting of ITGA6 for therapy purposes the readers are referred to read the review recently published by Zhang and collaborators (Zhang, 2023). “

  1. There are other emerging technologies like mRNA therapy or CRISPR-CAS9, to target such diseases, the authors need to discuss about the recent research findings and
    the ongoing trials.

We appreciate the suggestion, and regarding this, we want to point out that we have been citing research using these technologies all along the manuscript.  We also look for information about potential ongoing trials as requested, however, we could not find enough information.  Also, we want to address that our review is intended for the understanding of the role of ITGA6 in cancer biology.

  1. The authors can add a section on the ongoing clinical trials and the expected timeline of outcomes.

Thank you for the suggestion, however, we could not find enough information regarding ongoing clinical trials using itga6.  Nevertheless, we are pointing out a recently published review directed in this direction. (ZHANG, W., YE, J., LI, X., LI, Y. & FENG, G. 2023. Integrin α6 targeted cancer imaging and therapy. Vis Cancer Med, 4, 4).  This can be found on page 22 lines 15-16.

  1. The discussion is too short, can the authors discuss more broadly in the context of checkpoint inhibitors and their implications in cancer.

Thank you for the suggestion, however, we respectfully disagree with this comment.  We strongly believe that the information provided in this manuscript covers multiple relevant aspects of the regulation and function of itga6 in cancer biology.  Regarding the context of checkpoint inhibitors and their implications in cancer, we can find an angle that connects this with itga6, thus we decided not to pursue this suggestion.

  1. The references are in a different font size and the font size on page 15 needs formatting.

We have corrected this, thanks for bringing it to our attention.

  1. The authors described about contradictory studies on page 15 but need to refer to such articles as they are missing.

Thank you for pointing out this to us.  We added the corresponding reference, to the following sentence found on page 14, lines 24-26:

“Despite the high level of vascularization, anti-angiogenic medicines have failed to achieve the desired results [107-109].”

  1. The authors need to describe about the cross talk of ITGA6 with other signaling pathways, even though the authors briefly touched upon it.

We appreciate the comment of the reviewer, however, this manuscript is a review that points generally to the many aspects of ITGA6, and we can only imply it briefly.  The intention is that readers can refer to the interesting fields by the references presented. Describing everything in expanding information is not possible since we have limited space. Additionally, we presented the number of related studies as possible and suitable. 

Reviewer 2 Report

The manuscript by Khademi et al, entitled “Regulation and Functions of a6-Integrin (CD49f) in Cancer Biology” is well-written and provides an overview ofthe role of a6-Integrin in cancers. However, I have the following concerns as follows:

Comments

1.        Some references are not appropriate. It may be appropriate to include original papers instead of review papers in some references. For example, P3, line 3 (Beaulieu, Gang); P7, Line 24 (Li et al) and Line 26 (Su et al); P11, Line 33 (Zhou) and Line 46 (Cooper, Zhou).

2.        Please check all references through the text. For example, I should like to point out that the reference quoted as Beaulieu J. F. 2020 is in actually Beaulieu J. F. 2019. P3, Line 6 (Beaulieu 2020). Is the reference correct? I cannot find any description in the reference.

3.        There is no explanation about Figure 1C in the text. Figure 1C and Figure 2 is redundant. Can the authors include Figure 2 into Figure 1 or Figure 1C into Figure 2? P4, last sentence. Figure 2C should be Figure 2A.

4.        I recommend to additional subsection about miR in the section 2. 

5.        Some figure colors are so dark. For example, Figure 2B, M6A. Therefore, it is difficult to see the letter in the colors. 

6.        Section 2.1 is too long and lacks emphasis. It should be reduced in length.

7.        Please add the references into p5, line 1 from the bottom (The Src-family kinases……integrin beta subunits) and p6, line 1 (…….SFK in a FAK-independent way). 

8.        Please add the information on PI3K and Akt into P6, first paragraph and Figure 3.

9.        Section 3. Overview of Integrin Signaling. 1st and 2nd paragraphs. Do these description apply to all integrins?

10.  Figure 1B, 5B, and 5C are information about regulation of ITGA6 splicing variants. I would suggest an attempt to tie the information together. 

11.  Figure 5A. Is TGF-bTGF-bR? How does ITGA6 regulate TGF-b? Please add the explanation for it in Figure 5A.

12.  Figure 5B and 5C. Size-up of ITGA6-A or ITGA6-B is not appropriate to describe upregulation.

13.  p10, first paragraph to p11 second paragraph.. The description does not correspond to the subsection title, EMT.

14.  Section 7. Most of all are about lymphoblastic leukemia cells. How about other types of cancers?

Author Response

Reviewer 2 MDPI

  1. Some references are not appropriate. It may be appropriate to include original papers instead of review papers in some references. For example, P3, line 3 (Beaulieu, Gang);
    P7, Line 24 (Li et al) and Line 26 (Su et al); P11, Line 33 (Zhou) and Line 46 (Cooper, Zhou).
  2. Please check all references through the text. For example, I should like to point out that the reference quoted as Beaulieu J. F. 2020 is in actually Beaulieu J. F. 2019. P3, Line
    6 (Beaulieu 2020). Is the reference correct? I cannot find any description in the reference.

Dear reviewer, thank you so much for pointing out to us the mistakes in the references.  We have addressed this issue, and strongly believe that it has been taken care.

  1. There is no explanation about Figure 1C in the text. Figure 1C and Figure 2 is redundant. Can the authors include Figure 2 into Figure 1 or Figure 1C into Figure 2? P4, last sentence. Figure 2C should be Figure 2A.

Thank you for the suggestions.  Regarding the explanation or association about Figure 1C in the text, this can be found in section 2.3 Regulation of itga6 by microRNAs, which is on page 5 lines 44-49 and page 6 lines 1-4.

We agreed with your comment that Fig 1C and 2 are redundant. Thus, we removed Fig 1C, and now fig 2 is fully dedicated to mRNA regulation of itga6 by miRNAs.

  1. I recommend to additional subsection about miR in the section 2.

In our original submission, we already included a section of miRNA regulation (Section 2.3).  In this revised version we expanded it.  This can be found in pages 5-line 43 to 6-line 26.

  1. Some figure colors are so dark. For example, Figure 2B, M6A. Therefore, it is difficult to see the letter in the colors.

Thanks for pointing out this to us, we proceeded accordingly.

  1. Section 2.1 is too long and lacks emphasis. It should be reduced in length.

Several sentences were removed.

  1. Please add the references into p5, line 1 from the bottom (The Src-family kinases……integrin beta subunits) and p6, line 1 (…….SFK in a FAK-independent way).

Thanks for pointing out this, we have corrected it by adding the corresponding references. This can be found in page 7 lines 43-44.

  1. Please add the information on PI3K and Akt into P6, first paragraph and Figure 3.

We added the following and reference on page 8 lines 9-11.  “In leukemia, the enzyme PI3K plays a critical role in transducing extracellular signals that control cell proliferation, survival, and migration ([Yao et al., 2018]). “

The figure 3 was removed, as was not specific for ITGA6.

  1. Section 3. Overview of Integrin Signaling. 1 and 2 paragraphs. Do these description apply to all integrins?

Your interpretation is correct. 

  1. Figure 1B, 5B, and 5C are information about regulation of ITGA6 splicing variants. I would suggest an attempt to tie the information together.

We followed your suggestion, and in the text, we put together the information of ITGA^ regulation by splicing methods.  This can be found in section 2.2 Alternative splicing regulation by itga6 (from page 4 line 22 to page 5 line 41).  The two last paragraphs were moved on from later section to tie the information together. 

  1. Figure 5A. Is TGF-bTGF-bR? How does ITGA6 regulate TGF-b? Please add the explanation for it in Figure 5A.
  2. Figure 5B and 5C. Size-up of ITGA6-A or ITGA6-B is not appropriate to describe upregulation.

Thank you for pointing out this, we corrected it.

  1. p10, first paragraph to p11 second paragraph.. The description does not correspond to the subsection title, EMT.

Thank you for pointing out this, we agreed and moved those sentences to the Alternative splicing regulation of itga6, on page 5 lines 24-41.

  1. Section 7. Most of all are about lymphoblastic leukemia cells. How about other types of cancers?

Indeed, we are hematologists and biologists, and therefore, our main focus is on hematologic malignancies, however, we decided to consider other solid tumors as well during the rest of the manuscript. 

Round 2

Reviewer 1 Report

The article entitled 'Regulation and Functions of α6-Integrin (CD49f) in Cancer Biology' by Khademi et al is very interesting. 

No comments

Author Response

Thank you. 

Reviewer 2 Report

Comments
1.  The authors did not comply with my previous comments 11 and 12 (Figure 5A. Is TGF-beta TGF-betaR?; How does ITGA6 regulate TGF-beta? Please add the explanation for it in Figure 5A.).

2. In response 14. The authors said that they mainly focused on 
hematologic malignancies. However, there is no description in the text or title. First I expected that this review mainly focused on ITGA6 in solid tumors. Maybe other readers also feel the same thing. Therefore, I suggest that the authors state it in the title as well as the text. Since the text in the manuscript is too long, the authors maybe focus on hematologic malignancies by removing the description of other types of 
cancers.

3. Until now many studies have shown that beta4 integrin promotes tumor malignancies and metastasis. However, the authors describe that alpha6beta4 integrin shows opposing effects in malignancies and may act as an inhibitor of cancer metastasis by introducing few papers. Please check all papers about beta4 integrin in cancers and reconsider whether the statement is appropriate or not.

4. Section 5. The section title “ITGA6 and Cancer Metastasis” is not appropriate. Also the content in the introduction (tumor metastasis process) is different from the content in the following subsection (Hypoxia, angiogenesis, EV).

Minor editing of English language required.

Author Response

Comments and Suggestions for Authors

Comments
1.  The authors did not comply with my previous comments 11 and 12 (Figure 5A. Is TGF-beta TGF-betaR?; How does ITGA6 regulate TGF-beta? Please add the explanation for it in Figure 5A.).

I added a brief explanation to the legend of Fig. 4 (previously Fig. 5).  We apologize for missing that previous requirement, however, in the first resubmission we added the explanation to the main text. 

2. In response 14. The authors said that they mainly focused on 
hematologic malignancies. However, there is no description in the text or title. First I expected that this review mainly focused on ITGA6 in solid tumors. Maybe other readers also feel the same thing. Therefore, I suggest that the authors state it in the title as well as the text. Since the text in the manuscript is too long, the authors maybe focus on hematologic malignancies by removing the description of other types of cancers.

We apologize for the confusion that we created with our response to your initial comment.  It is true we are hematologists and therefore we put extra effort into blood malignancies.  However, this review is not intended exclusively for hematologic malignancies, and this is evident all along the manuscript.  However, it is true that we have a specific section for the role of ITGA6 in metastasis to the CNS in acute lymphoblastic leukemia. After this clarification, we don't think it is necessary to change the title or remove sections from the manuscript.  Finally, it was been always pointed out the intention of the review, for example in the abstract you can find out this sentence, which included solid and hematological malignancies:

“The importance of ITGA6 in the progression of a number of cancers including hematological malignancies suggests its potential usage as a novel prognostic or diagnostic marker and useful approach to therapeutic target for better clinical outcomes.”

,   

3. Until now many studies have shown that beta4 integrin promotes tumor malignancies and metastasis. However, the authors describe that alpha6beta4 integrin shows opposing effects in malignancies and may act as an inhibitor of cancer metastasis by introducing few papers. Please check all papers about beta4 integrin in cancers and reconsider whether the statement is appropriate or not.

We agree with the reviewer's comment that integrin b4 has been involved in metastasis and that there is plenty of published evidence supporting the role of b4 in metastasis.  We are not disputing this in any way or form. Now, we need to remember that integrin b4 is capable of heterodimer with multiple alpha subunits, and each heterodimer combination has a specific cell type function.  In the manuscript (page 14, lines 13-14), we are just referring to studies indicating that the integrin heterodimer of a6b4 can act as an inhibitor of metastasis, this does not deny the role of b4 alone in metastasis.

4. Section 5. The section title “ITGA6 and Cancer Metastasis” is not appropriate. Also the content in the introduction (tumor metastasis process) is different from the content in the following subsection (Hypoxia, angiogenesis, EV).

Thanks for pointing out this, and accordingly, we modified the subtitle to

‘Involvement of ITGA6 in Cancer Metastasis.”

In the first paragraph of this subsection, we intended to describe a general overview of the metastasis process, and this is followed up by specific sections (ITGA6 role in hypoxia, ITGA6 and angiogenesis, ITGA6 in extracellular vesicles and ITGA6 and Metastasis to the Central Nervous System (CNS) of Acute Lymphoblastic Leukemia (ALL) Patients), in which we delineate the role that ITGA6 plays in each of these important processes involved also in metastasis.  Thus, to make this clearer, we added the following sentences on page 14 lines 14-16: “The following sub-sections are intended to highlight the involvement of ITGA6 in multiple processes that support metastasis.”

Finally, our manuscript was edited in the English language as recommended. 

Round 3

Reviewer 2 Report

Comments

1.     Page 3, Line 25. The abbreviation of ECM is not correct.

2.     Page 4, Line 28-29. “However, ….and may act as an inhibitor of cancer metastasis” is not correct. The references 123,24 which are the references that authors referred, states that most studies have shown that beta4 integrin positively regulates tumor progression but some controversy exists. Therefore, this sentence (Line 28-40) is misleading.

Author Response

Dear reviewer, 

Thanks for your diligent revision of our manuscript.  Because of your comments and revisions, our manuscript has improved significantly. 

We corrected the ECM abbreviation that you indicated.

We also remove the sentence on page 4 lines 28-29 and its reference, as it was controversial and potentially misleading, as you indicated.

Thank you.   
